# Effects of the Co-Overexpression of the *BCL* and *BDNF* Genes on the Gamma-Aminobutyric Acid-Ergic Differentiation of Wharton’s-Jelly-Derived Mesenchymal Stem Cells

**DOI:** 10.3390/biomedicines11061751

**Published:** 2023-06-18

**Authors:** Paulina Borkowska, Julia Morys, Aleksandra Zielinska, Jan Kowalski

**Affiliations:** Department of Medical Genetics, Medical University of Silesia, 41-200 Sosnowiec, Poland

**Keywords:** mesenchymal stem cells, co-overexpression, *BCL2*, *BCLXL*, *BDNF*, GABA-ergic neurons

## Abstract

One of the problems with using MSCs (mesenchymal stem cells) to treat different neurodegenerative diseases of the central nervous system is their low ability to spontaneously differentiate into functional neurons. The aim of this study was to investigate how the co-overexpression of the *BCL* and *BDNF* genes affects the ability of genetically modified MSCs to differentiate into GABA-ergic neurons. A co-overexpression of two genes was performed, one of which, *BCL*, was supposed to increase the resistance of the cells to the toxic agents in the brain environment. The second one, *BDNF*, was supposed to direct the cells onto the neuronal differentiation pathway. As a result, the co-overexpression of both *BCL2* + *BDNF* and *BCLXL* + *BDNF* caused an increase in the *MAP2* gene expression level (a marker of the neuronal pathway) and the *SYP* gene that is associated with synaptogenesis. In both cases, approximately 18% of the genetically modified and then differentiated cells exhibited the presence of the GAD protein, which is characteristic of GABA-ergic neurons. Despite the presence of GAD, after both modifications, only the *BCL2* and *BDNF* co-overexpression correlated with the ability of the modified cells to release gamma-aminobutyric acid (GABA) after depolarization. Our study identified a novel model of genetically engineered MSCs that can be used as a tool to deliver the antiapoptotic proteins (BCL) and neurotrophic factor (BDNF) directly into the brain microenvironment. Additionally, in the investigated model, the genetically modified MSCs could easily differentiate into functional GABA-ergic neurons and, moreover, due to the secreted BCL and BDNF, promote endogenous neuronal growth and encourage synaptic connections between neurons.

## 1. Introduction

Genetically modified mesenchymal stem cells seem to be worth investigating in terms of their possible use in transplants in patients suffering from neurodegenerative diseases [1,2,3]. The basic features of cells, such as their multilineage potential [4,5], low tumorogenic potential and low level of MHC (major histocompatibility complex) protein expression [6], can be enriched by the features that are acquired from genetic modifications. The main problem inherent in using unmodified mesenchymal stem cells is their negligible survivability after the transplantation procedure [7,8,9]. Poor environmental conditions in the brain, such as insufficient nutrition and oxygen-free radical toxicity, limit the efficacy of BMSC therapy [10]. Another problem seems to be their poor ability to spontaneously differentiate, especially into the cells that originate from a germ layer other than themselves [11,12]. Thus, the hypothesis of the co-overexpression of genes was created, which has a chance to modify both of these features. To investigate this hypothesis, two genes were selected: one from the *BCL* (B-cell lymphoma) gene family, which was expected to increase cell survival and resistance to toxic factors, and the other—BDNF (brain-derived neurotrophic factor)—whose overexpression should increase their ability to differentiate into the progenitors of nerve cells or mature cells or different types of neurons. Previous studies have shown that the co-overexpression of the *BCL2* and *BDNF* or *BCLXL* and *BDNF* genes in an analogous transduction system increases cell survival under toxic conditions in an in vitro culture [13,14] and increases their ability to differentiate into cells that exhibit a cholinergic and/or dopaminergic phenotype [13].

Because neurodegenerative diseases are rarely caused by a deficit of one type of cells in one brain structure, and it seems ideal to optimize the genetic modification of MSCs in such a way that it would be possible to transplant cells that have a wide potential for differentiation [8]. Demonstrating the influence of the co-overexpression of *BCL2* and *BDNF* as well as *BCLXL* and *BDNF* on the possibility of obtaining several types of nerve cells in a culture would support the fact that the proposed modification scheme could be tested in vivo in various types of diseases because, as a result of the modification, cells with increased resistance to toxic conditions [13,14], which are also progenitors of nerve cells, would be obtained. Performing the in vitro transduction and then transplantation of freshly modified cells would subject them to the microenvironment of the brain, which would define their final path of differentiation depending on what cells are needed [15]. How the co-overexpression of *BCL2* and *BDNF* or *BCLXL* and *BDNF* genes affects the ability to differentiate into GABA-ergic cells remains unexplored and, therefore, this analysis was the aim of this study.

## 2. Materials and Methods

### 2.1. Isolating, Culturing and Characterizing the WJ-MSC

The study was approved by the Bioethical Committee of the Medical University of Silesia in Katowice (Resolution No. KNW/0022/KB/195/14). The participants (mothers) were informed of the procedure and gave their written consent for the use of the umbilical cords.

The WJ-MSCs (Wharton-Jelly-derived mesenchymal stem cells) were isolated from human umbilical cords, which were diced into fragments and placed on culture dishes containing a Mesenchymal Stem Cell Expansion Medium (1X) (#SCM015; Sigma Aldrich^®^, Merck^®^, St. Louis, MO, USA) that had been supplemented with a 1% Antibiotic Antimycotic Solution (#P06-07300; PAN-Biotech™, Am Gewerbepark, Aidenbach, Germany). A seven-day expansion was performed in a humidified incubator at 37 °C with 5% CO_2_. During the process, the WJ-MSCs migrated from the tissue fragments to the bottom of the culture dishes. After seven days, the tissue fragments were removed. The WJ-MSCs were passaged for the first time and put into the culture flasks. The cells were treated with 0.05% trypsin with 0.02% EDTA in phosphate-buffered saline (PBS) without calcium and magnesium (#PCS-999-003; ATCC^®^, University Blvd, Manassas, VA, USA) for three min. in order to form single cells. After that, to perform a rapid inactivation, a trypsin neutralizing solution (5% FBS (fetal bovine serum) in phosphate-buffered saline without calcium and magnesium) (#PCS-999-004; ATCC^®^, University Blvd, Manassas, VA, USA) was used. The volume of trypsin and trypsin inactivator was used in a 1:1 ratio. During the entire experiment or until banking, the cells were cultured in a mesenchymal stem cell basal medium (#PCS-500-030; ATCC^®^, University Blvd, Manassas, VA, USA) that had been supplemented with a mesenchymal stem cell growth kit for adipose- and umbilical-derived MSC-low serum (#PCS-500-040; ATCC^®^, University Blvd, Manassas, VA, USA). The final concentration for each component in a complete mesenchymal stem cell growth medium was as follows: rh FGF basic: 5 ng/mL; rh FGF acidic: 5 ng/mL; rh EGF: 5 ng/mL; fetal bovine serum: 2% and L-Alanyl-L-Glutamine: 2.4 mM. Then, the cells at an 80% confluence were passaged and cryopreserved to be left as a WJ-MSC bank for further use.

The homogeneity of the WJ-MSCs was quantified using flow cytometry; the presence of CD73, CD90, CD34, CD11b, CD19, CD45 and HLA-DR (human leukocyte antigen—DR isotype) was also determined. The samples were prepared by trypsinizing the cells with 0.05% trypsin/EDTA (#P10-029500; PAN-Biotech™, Am Gewerbepark, Aidenbach, Germany) and suspending them in phosphate-buffered saline (DPBS) (#P04-36500; PAN-Biotech™, Am Gewerbepark, Aid-enbach, Germany). The suspensions prepared in this way were then analyzed via flow cytometry (BD FACSAria II, Qume Dr., San Jose, CA, USA; BD FACSDiva Software V6.1.2), seeking the areal markers presented by the WJ-MSCs [16]. The WJ-MSCs were also characterized based on their differentiation capacity toward adipocytes and osteocytes. For adipogenic and osteogenic differentiation, the following conditions were provided: a culture of WJ-MSCs in StemMACS AdipoDiff Media (#130-091-677; Miltenyi Biotec© Teterow, Germany) for over 21 days and detection with the use of Oil Red O (#O9755-256; Sigma Aldrich^®^, Merck^®^, St. Louis, MO, USA) for adipocytes, and a cell culture in StemMACS OsteoDiff Media (#130-091-678; Miltenyi Biotec© Teterow, Germany) for the same time period and detection with the use of alkaline phosphatase using a 5-bromo-4-chloro-3-indolyl phosphate/Nitro blue tetrazolium (BCIP/NBT) substrate (#B5655-5TAB; Sigma Aldrich^®^, Merck^®^, St. Louis, MO, USA) for osteocytes. All of the experimental measures taken with the mentioned reagents were conducted according to the instructions provided by the manufacturer. The control probe was not treated with any additional factors and consisted of the WJ-MSCs cultured in DMEM/F12 medium (#P04-41250; PAN-Biotech™, Am Gewerbepark, Aidenbach, Germany) supplemented with 15% FBS (#P30-8500; PAN-Biotech™, Am Gewerbepark, Aidenbach, Germany) and 1% Antibiotic Solution [16].

### 2.2. Lentiviral Transduction and Neuronal Differentiation

The lentivirus production protocol started with culturing the HEK 293T cells in 25 cm^3^ culture flasks in a proper medium that consisted of DMEM (#P04-05550; PAN-Biotech™, Am Gewerbepark, Aidenbach, Germany) medium supplemented with 10% FBS (#P04-03590; PAN-Biotech™, Am Gewerbepark, Aidenbach, Germany) and 1% Antibiotic Solution (#P06-07300; PAN-Biotech™, Am Gewerbepark, Aidenbach, Germany) at 37 °C in 5% CO_2_ conditions. After achieving the 80% confluence in culture flasks, the cells were trypsinized and transferred onto 10 cm Ø Petri dishes for culture under humidified conditions, as mentioned previously. Subsequently, the 20 µg of the respective plasmid (Figure 1) plus 6 µg of pMD2.G (pMD2.G from Didier Trono (Addgene plasmid #12259; http://n2t.net/addgene:12259, accessed on 28 May 2023; RRID:Addgene_12259) and 15 µg psPAX (psPAX2 from Didier Trono (Addgene plasmid #12260; http://n2t.net/addgene:12260, accessed on 28 May 2023; RRID:Addgene_12260) plasmids were added. The supernatants containing lentiviral vectors were collected from each Petri dish with a sterile Pasteur pipette, infiltrated with the use of Corning^®^ 150 mL Vacuum Filter/Storage Bottle System (#431153; Corning^®^, Glendale, AZ, USA) and diluted using the Lenti-X Concentrator (#631231; TaKaRa Bio©; Kyoto, Japan). Centrifuged lentiviruses were aliquoted and stored at −70 °C. These stocks were optimized on MSC cultures by performing the transduction and checking the amount of positively transduced cells with the use of flow cytometry. The coding sequences of *BCL2* and *BCLXL* were cloned to LeGO-iG2 (Figure 1a), and the coding sequence of *BDNF* was cloned to LeGO-iT2 backbones (Figure 1b). As a result, the plasmids were obtained as follows: LeGO-iG2-Bcl-2 (Figure 1c), LeGO-iG2-Bcl-XL (Figure 1d) and LeGO-iT2-BDNF (Figure 1e). Boris Fehse kindly provided the LeGO-iG2 (Addgene plasmid # 27341; http://n2t.net/addgene:27341, accessed on 28 May 2023; RRID:Addgene_27341) and LeGO-iT2 (Addgene plasmid # 27343; http://n2t.net/addgene:27343, accessed on 28 May 2023; RRID:Addgene_27343) [17].

The day before the transduction, 100,000 WJ-MSCs per 35 mm Ø dish were plated in order to achieve a 70–80% confluence immediately prior to the transduction. An 18 h incubation of the preconfluent cells with an optimal dilution of a virus in an Opti-MEM I Reduced Serum Medium (#31985070; Gibco^®^, Thermo Fischer Scientific^®^, Grand Island, NE, USA) in the presence of 5 μg/mL polybrene (without any antibiotics) was followed by a 12-day neuronal differentiation with a medium containing Neurobasal PLUS (#A3582901; Gibco^®^, Thermo Fischer Scientific^®^, Grand Island, NE, USA) and a B27 PLUS supplement (#A3582801; Gibco^®^, Thermo Fischer Scientific^®^, Grand Island, NE, USA).

### 2.3. RT-qPCR

The differentiated cells were used to extract the total cellular RNA using NucleoZOL (#740404.200; Macherey-Nagel GmbH & Co. KG, Richterstr., Berlin, Germany) according to the manufacturer’s protocol. Spectrophotometry was used to determine the quality and concentration of the RNA. The primers that were used for the selected genes were purchased from Sigma Aldrich (#KiCqStart SYBR Green Primers; Sigma Aldrich^®^, Merck^®^, St. Louis, MO, USA). *RPS17* was used as the reference gene. The sequences of the primers that were used were as follows: *RPS17* (P_F_ 5′ CCATTATCCCCAGCAAAAAG 3′; P_R_ 5′ GAGACCTCAGGAACATAATTG 3′; Primer Pair ID H_RPS17_1), *SYP* (P_F_ 5′ CCCTTCGGTATTGTTCAAAG 3′; P_R_ 5′ TTTGACTAGGTGGTTAAGGAG 3′; Primer Pair ID H_SYP_1), *GAD* (P_F_ 5′ GGTTTGGATATGGTTGGATTAG 3′; P_R_ 5′ GGAGCAATTTCATAGGTGAAC 3′; Primer Pair ID H_GAD_2), *BAX* (P_F_ 5′ AACTGGACAGTAACATGGAG 3′; P_R_ 5′ TTGCTGGCAAAGTAGAAAAG 3′; *MAP2* (P_F_ 5′ GAAGATTTACTTACAGCCTCG 3′; P_R_ 5′ GGTAAGTTTTAGTTGTCTCTGG 3′; Primer Pair ID H__MAP2_2). A GoTaq 1-Step RT-qPCR System (#A6020; Promega©, Woods Hollow Rd., Madison, WI, USA) was used to perform the one-step RT-qPCR in triplicate. A 10 μL reaction with 15 ng of total RNA and a 0.2 μM final primer concentration for each forward and reverse primer was used to perform the RT-qPCR in a C1000 Touch Thermal Cycler equipped with a CFX96 Real-Time System (Bio-Rad CFX Manager, Alfred Nobel Dr., Hercules, CA, USA; Software Version 3.1). The RT-qPCR procedure comprised a 15 min RT reaction at 37 °C, a 10 min PCR activation at 95 °C and then 40 cycles of a 10 s denaturation at 95 °C, a 30 s annealing at 60 °C at the melting temperature of the lowest primers’ pair and a 30 s PCR extension at 72 °C. Lastly, a melting curve analysis was performed in order to confirm the RT-qPCR specificity. Each run of the RT-qPCR included negative controls that had no total RNA. A Bio-Rad CFX96 Real-Time System (Bio-Rad CFX Manager, Alfred Nobel Dr., Hercules, CA, USA; Software Version 3.1) was used to automatically determine the Ct value.

### 2.4. Evaluating the GAD67 Protein Expression

Indirect labeling with a specific anti-GAD67 antibody was used to identify the GAD67 protein-positive cells. After the differentiation, the cells were washed twice in PBS and fixed in PBS with 4% paraformaldehyde (#158127; Sigma Aldrich^®^, Merck^®^, St. Louis, MO, USA). The cells were then washed three times in PBS with 1% BSA (#A9418; Sigma Aldrich^®^, Merck^®^, St. Louis, MO, USA), permeabilized with 0.3% Triton X-100 (#T8787; Sigma Aldrich^®^, Merck^®^, St. Louis, MO, USA)/1% BSA/PBS with 10% normal goat serum (#005-000-121; Jackson ImmunoResearch©, St. Thomas’ Place, Ely, UK) for 45 min and incubated with rabbit anti-GAD67 antibody (dilution 1:250; # GTX101881 GeneTex©, Alton Pkwy., Irvine, CA, USA) in 1% BSA/PBS with 10% goat serum overnight at 4 °C. The cells were washed three times in PBS with 1% BSA and incubated with goat anti-rabbit IgG secondary antibody (DY405) (dilution 1:1000; # LS-C355899; LifeSpan BioSciences©, Shaker Rd., Suites Shirley, MA, USA) in 1% BSA/PBS for 1 h in the dark at room temperature. In order to determine any non-specific binding, similar staining was performed using a rabbit IgG isotype control (dilution 1:625; # bs-0295P; Bioss Antibodies©, Tradecenter Ste., Woburn, MA, USA). Flow cytometry (BD FACSAria II, Qume Dr., San Jose, CA, USA; BD FACSDiva Software V6.1.2) and fluorescence microscope (BX60, Olympus^®^, Shinjuku, Tokyo, Japan; cellSense Standard Software V3.2) were used to analyze the staining.

### 2.5. Gamma-Aminobutyric Acid Release Analysis

The procedure used to perform the assay was previously published [13]. The neurotransmitter levels were obtained and quantified using an ELISA (enzyme-linked immunosorbent assay) kit (#CEA900Ge; Cloud-Clone Corp.©, W. Fernhurst Dr., Katy, TX, USA) according to the manufacturer’s instructions. An ELX 800 IU automated Microplate Reader (Bio-Tek Instruments; Gene 5 Software V 3.02) was used to read the absorbance at 450 nm. A quadratic log-log curve fit was used to analyze the results. The supernatants (low and high K^+^ = before and after the GABA release) were analyzed for each well. In order to minimize the effect of the differences in the number of cells in the wells, the amount of GABA that had been produced per well was calculated based on the difference between the amount of GABA in a high K^+^ supernatant and the amount of GABA in a low K^+^ supernatant.

### 2.6. Statistical Analysis

All of the collected data were compiled using Microsoft^®^ Excel 2023 V16.70 spreadsheets (Microsoft Corporation, Redmond, WA, USA) and statistically analyzed using GraphPad^®^ Prism 8.0 software (Graphpad Software Inc., San Diego, CA, USA). First, a column analysis (Shapiro–Wilk normality test) was performed to determine whether there was a Gaussian distribution of the data. Next, the data were analyzed using the ordinary one-way ANOVA test followed by a Tukey multiple comparison test. For all of the tests, *p* < 0.05 was considered to be statistically significant. Reproducible results were obtained from six or more samples. The data are presented as the mean ± SD. The qRT-PCR results are presented as the fold change (2^−ΔΔCT^).

## 3. Results

### 3.1. The Co-Overexpression of BCL and BDNF Affects the Expression Level of the Neuronal Pathway Genes

The investigated co-overexpression of BCL and BDNF resulted in an increase in the gene expression level of the neuronal pathway genes in the culture of genetically modified MSCs. It was observed that the co-overexpression of BCL2 and BDNF caused an increase in the expression of *MAP2* (Figure 2a, iG2 + iT2 vs. BCL2 + BDNF) and *GAD* (Figure 2c, iG2 + iT2 vs. BCL2 + BDNF: *p* < 0.01) and a ten-fold increase in the *SYP* gene level (Figure 2b, iG2 + iT2 vs. BCL2 + BDNF: *p* < 0.01). At the same time, it was observed that the co-overexpression of BCL2 and BDNF changed the *BAX* gene expression level in accordance with the control group (Figure 2d, Control vs. BCL2 + BDNF: *p* < 0.001). In turn, because of the co-overexpression of BCLXL and BDNF, the differences between the investigated groups were observed in accordance with *GAD* and *BAX* expression levels (Figure 2c,d; Control vs. BCLXL + BDNF: *p* < 0.05).

### 3.2. The Co-Overexpression of BCL and BDNF Affects the GAD67 Protein Expression Level

The investigated co-overexpression of BCL and BDNF resulted in an increase in the GAD67 protein expression level in the culture of genetically modified MSC. It was observed that both the co-overexpression of BCL2 and BDNF (Figure 3d vs. Figure 3f; Figure 4b vs. Figure 4c and Figure 5, iG2 + iT2 vs. BCL2 + BDNF: *p* < 0.01) and the co-overexpression of BCLXL and BDNF (Figure 3d vs. Figure 3h; Figure 4b vs. Figure 4d and Figure 5, iG2 + iT2 vs. BCLXL + BDNF: *p* < 0.001) increased the expression of the GAD67 protein. 

In the cultures with an increased expression of the BCL2 and BDNF, 16% of the cells, and in cultures with an increased expression of the BCLXL and BDNF, 18% of the cells expressed the GAD67 protein that is characteristic of the GABA-ergic neurons, while in the control samples, only 4%, and in samples that had been transduced with empty vectors, only 6% of the cells exhibited the presence of GAD67. This result was confirmed using two independent methods: flow cytometry and immunofluorescence. A total of 10,000 cells were counted and analyzed.

### 3.3. The Co-Overexpression of BCL and BDNF Affects the Release of Gamma-Aminobutyric Acid

It was demonstrated that only the overexpression of the BCL2 and BDNF increased the release of gamma-aminobutyric acid as a result of depolarization. In the samples with an overexpression of the BCL2 and BDNF, after the differentiation process, the basic level of the release of GABA within 30 min. was 3.64 pg/mL ± 0.76. As a result, a 30 min. depolarization increased the level of the GABA release to 3.87 pg/mL ± 0.15. The amount of neurotransmitters that were produced increased by 0.23 pg/mL. For all of the other groups, including the cells with an overexpression of the BCLXL and BDNF, there was no increase in the GABA secretion as a result of depolarization.

## 4. Discussion

Huntington’s disease (HD) is an inherited neurodegenerative genetic disorder that leads to the onset of motor, neuropsychiatric and cognitive disturbances. HD is characterized by the loss of GABA-ergic neurons. In order to use genetically modified MSCs as a cell therapy in HD, it seemed to be necessary to investigate the effect of the gene co-overexpression on the ability of MSCs to differentiate into GABA-ergic neurons. In previous studies, the effect of the co-overexpression of the *BCL2* and *BDNF* genes on the ability of MSC to differentiate into cholinergic neurons was demonstrated [13]. It was also shown that the co-overexpression of both *BCL2* + *BDNF* and *BCLXL* + *BDNF* increases the ability of cells to survive under toxic conditions [13,14]. This confirmed that the increased viability of genetically modified MSCs is a significant advantage in the context of using these cells to treat HD. The role of ferroptosis in the etiopathogenesis of neurodegenerative diseases, such as HD, Parkinson’s disease and Alzheimer’s disease, is well known [18]. There are reports that stress the role of oxidative damage in the etiology of degeneration and neuronal damage in patients with HD [19]. The increased survival rate of genetically modified MSCs, generated via the co-overexpression of *BCL2* + *BDNF* and *BCLXL* + *BDNF*, increases the likelihood of transplant survival after a stroke procedure, such as transplantation. It is also important that the environment into which the transplant is introduced is a toxic environment with an increased amount of free radicals, with β-catenin in its phosphorylated form [20] and with factors that direct cells onto the ferroptosis path. Under such conditions, the demonstrated increased ability to survive under toxic conditions would be extremely desirable. Because it is important that the increased survival is not a permanent state not being subjected to control mechanisms, the increased level of the *BAX* gene expression that was demonstrated enabled us to conclude that the mechanisms that regulate cell death were not permanently silenced.

*MAP2* is an early neuronal marker that is commonly used to indicate these cells’ ability to differentiate into neurons. The generally higher expression of the mature neuronal marker MAP2 compared with the glial markers at the mRNA and protein levels suggest an enhanced potential for MSC neuronal differentiation [21]. In our study, we observed that especially the co-overexpression of *BCL2* and *BDNF* increased the *MAP2* gene expression level. Other authors have proved that the induction of the neural markers GFAP, NF, MAP2, and O4 and a decrease in the expression of the immature neural markers β-tubulin III, Nestin and NG2, co-exists with an increase in the level of TrkB, which is a BDNF receptor and occurs during the neural differentiation [22]. Another study provided evidence that *MAP2* in culture was upregulated by the neurotrophic factor NT3 [23] whose role in the central nervous system is similar to the role of BDNF.

Forming a network with neighboring cells is absolutely crucial for neuron-differentiated cells. These morphological and immunocytological features have been demonstrated. One of many genes that are involved in synaptogenesis is the *SYP* gene, which codes synaptophysin protein. We observed that the co-overexpression of combined *BCL2* + *BDNF* enhanced the *SYP* gene expression level. A similar effect in which the *SYP* expression level was also increased, along with increasing *MAP2*, was previously observed by other authors [24]. Moreover, an increased amount of BDNF in the brain environment is directly connected with an increased level of tyrosine kinase receptor B (trkB)—a receptor for BDNF. Other authors investigated whether signaling through tyrosine kinase receptor B influences neuronal survival, differentiation and synaptogenesis [25]. The overexpression of the *BDNF* gene may explain the increase in the expression of not only the *SYP* and *MAP2* genes but also the increase in the expression exhibited by the genes characterizing certain types of mature neurons, which was demonstrated in previous experiments using the same type of genetic modification. *MAP2*-positive cells could express different mature-neuronal markers such as *CHAT*a marker of cholinergic neurons; *TH*, a marker of dopaminergic neurons; or *TPH1*, a marker of serotoninergic neurons. The confirmed, increased expression of *MAP2* and *SYP* provided the basis for investigating whether the co-overexpression of the *BCL* and *BDNF* affects the *GAD67* gene and its protein expression level, which is a marker of GABA-ergic neurons. We observed that the co-overexpression of *BCL* and *BDNF* increased the protein expression level. In one recent study, the authors administrated dental pulp stem cells intravenously in an HD rat model. They demonstrated the stem cells homed in the striatum, cortex and subventricular zone using specific markers for human cells. Thirty days after the administration, an immunohistochemistry quantity analysis revealed a significant increase in the amount of BDNF, which proves its important role in GABA-ergic differentiation [26]. Other authors identified the GABA-ergic synapse pathway as being one of the major targets of the MSC-activated genes [27]. The enhancement of the GABA synapses was impaired by treatment with a Trk/neurotrophin receptors blocker and by TrkB receptors, thereby suggesting the involvement of BDNF as a mediator of such effect [28].

When investigating a model of differentiation, the co-overexpression of *BCL2* and *BDNF* seems to be especially important because of the correlation of the GAD-positive cells with the release of gamma-aminobutyric acid as a result of depolarization. Another author investigated this phenomenon in several central nervous system areas and demonstrated that BDNF can modulate the GABA-ergic synapses by regulating transmitter release [29]. A facilitatory action of BDNF on GABA release was previously reported by Pezet et al. [30], who demonstrated that an acute application of BDNF caused an increase in the KCl-elicited GABA release in a supernatant that had been collected from the isolated spinal cord in adult rats.

Since we have established that MSCs differentiate into GABA-ergic neurons, the evaluated cell therapy might be a great chance for patients suffering from HD to receive a promising treatment. However, the other types of neurons that various researchers obtained remain an auspicious treatment strategy in other neurodegenerative disorders as well. What has been proved by Venkatesh and Sen [31] is that MSCs may also differentiate into dopaminergic neurons, which could be potentially used as a cell therapy in Parkinson’s disease. Moreover, Ranjbaran et al. [32] conducted research in which it was stated that the MSCs may be a useful measure in treating anxiety regarding septic patients with encephalopathy by reducing the inflammation process and changing serotoninergic neurotransmission. Furthermore, Shwartz et al. [33] proposed a unique stem cell depression treatment by differentiating MSCs into cells exhibiting increased levels of functional glutamate transporters. In this case, it was proved that this type of MSC induces glutamatergic neurotransmission in the hippocampus, which leads to a meaningful improvement in the therapy used for the mentioned disorder.

Our study identified a novel model of genetically engineered MSCs as a tool for delivering antiapoptotic proteins (*BCL*) and neurotrophic factor (*BDNF*) directly into the brain microenvironment. Additionally, in the investigated model, the genetically modified MSCs could easily differentiate into functional GABA-ergic neurons and, therefore, due to the secreted *BCL* and *BDNF,* promote endogenous neuronal growth, encourage the formation of synaptic connections in damaged neurons, decrease apoptosis and, according to the literature, reduce the levels of free radicals [34] and regulate inflammation [24].

## 5. Conclusions

Based on the results of this study, it can be concluded that the co-overexpression of both *BCL* and *BDNF* can increase the ability of MSC to differentiate into GAD-positive cells and that the co-overexpression of *BCL2* and *BDNF* seems to be especially important as it correlates with the ability of the modified cells to release gamma-aminobutyric acid as a result of depolarization.

## Figures and Tables

**Figure 1 biomedicines-11-01751-f001:**
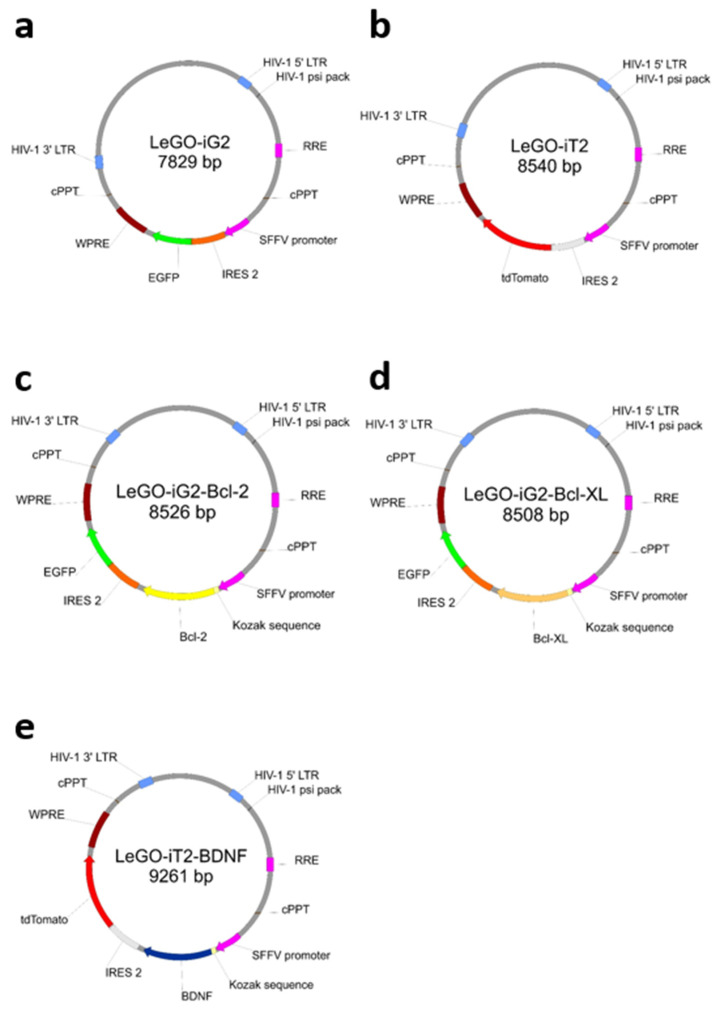
The simplified plasmid maps that were used to prepare the lentivirus. The lentiviral backbone plasmids contained the coding sequence of green fluorescence protein (*EGFP*) (**a**) or coding sequence of red fluorescence protein (*tdTomato*) (**b**). Into backbone “a”, coding sequence of *BCL2* gene (**c**) or *BCLXL* gene (**d**), and into backbone “b”, coding sequence of *BDNF* gene (**e**), were cloned to overproduced BCL2, BCLXL or BDNF proteins.

**Figure 2 biomedicines-11-01751-f002:**
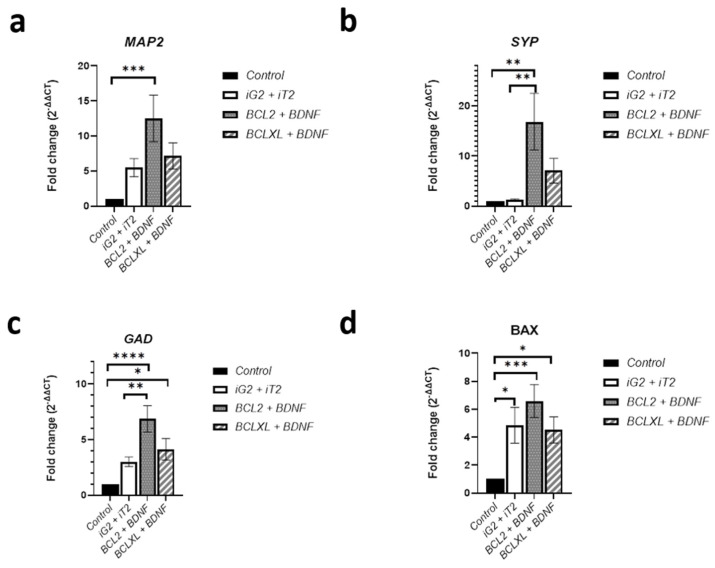
The qRT-PCR analysis is presented as the fold change (2^−ΔΔCT^) in the level of *MAP2* (**a**); *SYP* (**b**); *GAD* (**c**) and *BAX* (**d**) expression, which was normalized to the *RPS17* reference gene. The points represent the mean value ± SD (*n* = 9; three independent experiments). Statistically significant * *p* < 0.05; ** *p* < 0.01; *** *p* < 0.01; **** *p* < 0.0001; one-way ANOVA test followed by a post hoc Tukey’s test.

**Figure 3 biomedicines-11-01751-f003:**
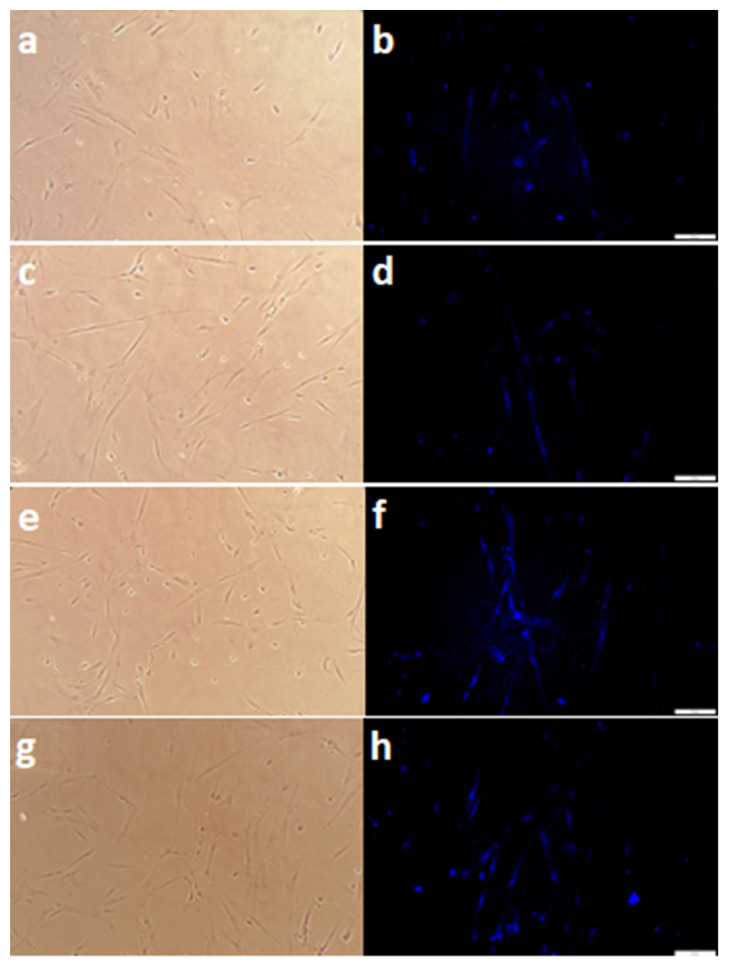
GABA-ergic neurons after co-overexpression and 12 days in differentiation medium. The GAD67 positive cells (blue) had a neuron-typical spindle shape. (**a**,**b**) Control. (**c**,**d**) iG2 and iT2 (Empty vectors) co-overexpression (transduction with vectors presented in Figure 1a,b). (**e**,**f**) BCL2 and BDNF co-overexpression (transduction with vectors presented in Figure 1c,e). (**g**,**h**) the BCLXL and BDNF co-overexpression (transduction with vectors presented in Figure 1d,e).

**Figure 4 biomedicines-11-01751-f004:**
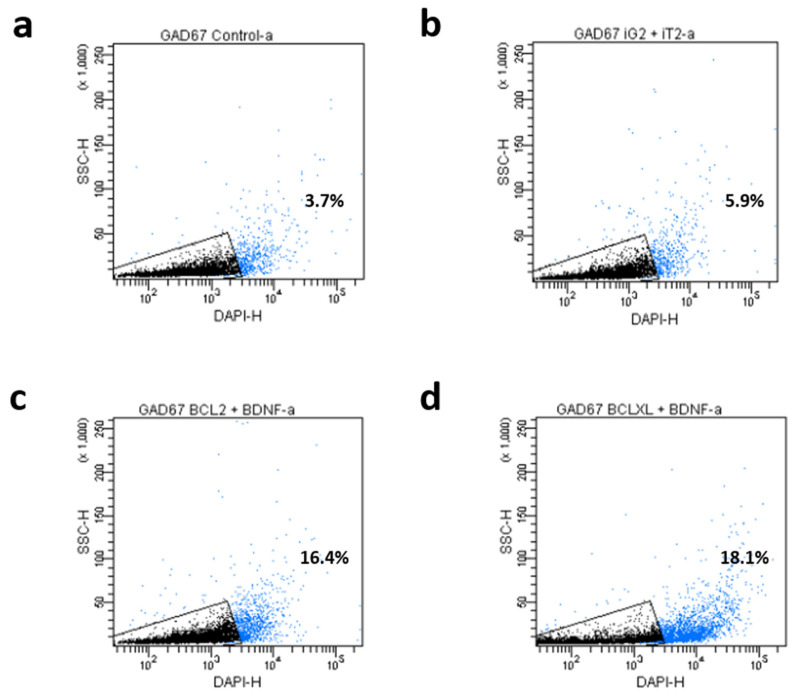
Flow cytometric analysis for the expression of GABA-ergic neural-specific proteins (GAD67) after 12 days of differentiation. The percentage values represent the mean value (*n* = 6; three independent experiments); (**a**) Control; (**b**) iG2 and iT2 (Empty vectors) co-overexpression (transduction with vectors presented in Figure 1a,b); (**c**) BCL2 and BDNF co-overexpression (transduction with vectors presented in Figure 1c,e); (**d**) BCLXL and BDNF co-overexpression (transduction with vectors presented in Figure 1d,e). A total of 10,000 cells were counted and analyzed.

**Figure 5 biomedicines-11-01751-f005:**
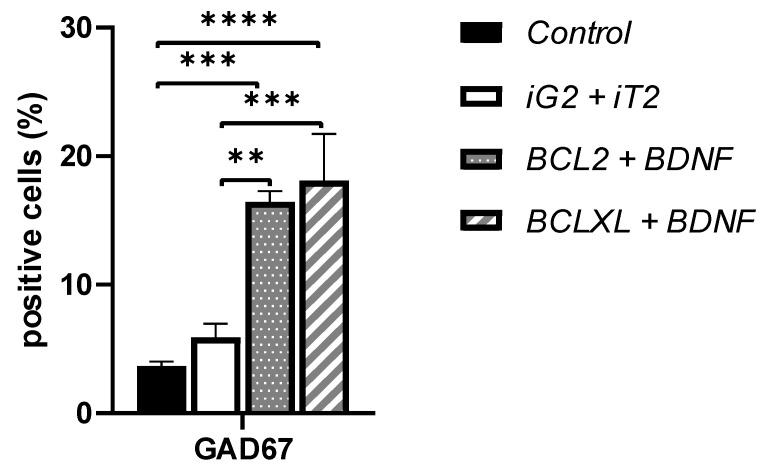
The number of GAD67 positive cells. The calculations were based on both flow cytometry analysis and immunofluorescence. A total of 10,000 cells were counted and analyzed. The points represent the mean value ± SD (*n* = 6; three independent experiments). Statistically significant ** *p* < 0.01; *** *p* < 0.01; **** *p* < 0.0001 one-way ANOVA test followed by a post hoc Tukey’s test.

## Data Availability

The datasets that were used and/or analyzed during the presented study are available from the corresponding author upon reasonable request.

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
