# Peer review of "Effects of the Co-Overexpression of the *BCL* and *BDNF* Genes on the Gamma-Aminobutyric Acid-Ergic Differentiation of Wharton’s-Jelly-Derived Mesenchymal Stem Cells"

_biomedicines, 2023, doi:10.3390/biomedicines11061751_

Round 1

Reviewer 1 Report

The aim of this study was to investigate how the co-overexpression of the BCL and BDNF genes affects the ability of genetically modifying MSC to differentiate into GABA-ergic neurons. They concluded that the co-overexpression of both BCL and BDNF can increase the ability of MSC to differentiate into GAD positive cells seems to be especially important for the modified cells to release GABA as a result of depolarization. The rationale appears reasonable. However, it may have been blunt below are some concerns:

1.      For the anti-apoptotic Bcl proteins, such as Bcl-2, Bcl-xL and MCL1, in general, Bcl-2 and Bcl-xL are functional similarity, why not use MCL1 as more control?

2.      For neurotrophic factor, BCL2 + BDNF and BCLXL (MCL1 as another control?), only alone may not be sufficient. How about other neurotrophic factor, such as PDGF or IGF1 or EGF?

3.      How BCL2 + BDNF specific to differentiate into GABA-ergic neurons (18%)? This need compare and discuss more other type neurons, such as Glutamat and Dopamine, need more discussion

Minor:

1.      gamma-aminobutyric acid (GABA) when first appears

2.      “a synaptic connections” delete “a”

Reviewer 2 Report

The authors present an interesting study examining the influence of BCL and BDNF genes in the differentiation of mesenchymal stem cells into functional neurons. Briefly, the authors highlight the difficulties in controlling cell phenotype upon application of mesenchymal stem cells within the nervous system. The uncertainty around their differentiation into cell types with little to no benefit in the context of the injury has called for methods in which the differentiated state could be controlled, with genetic methods leading such efforts. In this article, the authors induce the expression of BCL and BDNF through lentiviral means in an effort to illicit activity that promotes differentiation into neuronal cell types. The resultant efforts demonstrate a method which promoted the culture of GABA-ergic neurons, which would have application in treating a number of neurological disorders.

In reviewing the manuscript, I made a number of observations. The following should be considered by the authors when preparing a revision.

1.       There are several typos and grammatical errors within the text. These are typically minor in nature, but there are several and often distracting. The authors should review the manuscript and eliminate these instances to bring the writing to publication standard.

2.       There are instances where the authors allude to protocols that are described elsewhere. It would be preferable if the authors expanded upon these, if even as supplementary methods, such that readers have an opportunity to review the method and employ it as per this paper.

3.       Are there any data to validate the lentiviral effects on BCL and BDNF gene expression?

4.       Were the primers that were utilised for the study MQIE guidelines compliant?

5.       The resolution of the images needs to be improved. They are quite blurry in some instances, and could be scaled up to assist with this aspect.

6.       If possible, the contrast in the microscopy images should be improved such that the cell number and staiing are more obvious.

7.       I would recommend opening the methods section with the information on the ethics provided for this study before detailing the actual procedures involved.  

8.       The abbreviation of WJ-MSC should be expanded upon within the text for clarity.  

As mentioned elsewhere in my report, the writing warrants improvement in terms of typos and grammar. These are relatively minor in nature, but cumulatively amount to substantive changes. 

Round 2

Reviewer 2 Report

The authors have suitably addressed my comments.